# Association between Bone Mineral Density and Fat Mass Independent of Lean Mass and Physical Activity in Women Aged 75 or Older

**DOI:** 10.3390/nu13061994

**Published:** 2021-06-10

**Authors:** Marie Mathieu, Pascale Guillot, Typhaine Riaudel, Anne-Sophie Boureau, Guillaume Chapelet, Céline Brouessard, Laure de Decker, Gilles Berrut

**Affiliations:** 1Department of Geriatrics, Nantes University Hospital, 44093 Nantes, France; typhaine.riaudel@chu-nantes.fr (T.R.); annesophie.boureau@chu-nantes.fr (A.-S.B.); guillaume.chapelet@chu-nantes.fr (G.C.); celine.brouessard@chu-nantes.fr (C.B.); laure.dedecker@chu-nantes.fr (L.d.D.); gilles.berrut@chu-nantes.fr (G.B.); 2Department of Rheumatology, Nantes University Hospital, 44093 Nantes, France; pascale.guillot@chu-nantes.fr

**Keywords:** bone mineral density, lean mass, fat mass, osteoporosis, physical activity level, sarcopenia, osteosarcopenia, elderly women

## Abstract

(1) Osteoporosis and sarcopenia are frequent pathologies among the geriatric population. The interlink between these two diseases is supported by their common pathophysiology. The aim is to explore the relationship between bone mineral density (BMD) and body composition in women aged 75 or older. (2) From January 2016 to December 2019, women aged 75 or older of Caucasian ethnicity, who were addressed to perform a biphoton absorptiometry (DXA), were included in this observational study. Femoral neck T-score, lean mass, fat mass, and physical performances were measured. (3) The mean age of 101 patients included was 84.8 (±4.9) years old. Osteoporosis was present in 72% of patients. According to EWGSOP criteria, 37% of patients were sarcopenic. Osteosarcopenia was present in 34% of patients. The femoral neck T-score was significantly associated with fat mass (β = 0.02, 95% CI (0.01; 0.03), *p* < 0.05) in multivariable analysis. Osteosarcopenic patients had significantly lower fat mass (16.2 kg (±6.8) vs. 23.1 kg (±10.8), *p* < 0.001) and body mass index (BMI) (20.7 kg/m^2^ (±2.8) vs. 26.7 kg/m^2^ (±5.6), *p* < 0.001). (4) In postmenopausal women, fat mass is estimated to provide hormonal protection. While osteosarcopenia is described as a lipotoxic disease, fat mass and BMI would appear to protect against the risk of osteosarcopenia. This raises questions about the relevance of BMI and DXA.

## 1. Introduction

The human body, in its densities, has three components: bone mass, lean mass (LM), and fat mass (FM). During aging (especially if there is malnutrition), a dissociation between LM and FM is observed. These body modifications have implications on bone tissue, which is decreasing at the same time [1].

The decrease in bone mineral density (BMD), called osteoporosis, is traditionally defined by the measurement of BMD in biphoton absorptiometry or dual-energy X-ray absorptiometry (DXA) [2,3]. Associated with the decrease in BMD, an aging of the musculoskeletal system is observed and leads to a decrease in functional performances. The progressive and generalized decrease in mass, strength, and muscle function defines sarcopenia [4]. It is estimated that, between 20 and 80 years, skeletal muscles lose 50% of their weight [5], leading to a significant increase of sarcopenia in the elderly population [6]. The most affected muscle fibers are those involved in rapid muscle responses and fine movements, resulting in a loss of strength and power with a functional impact [7]. Sarcopenia increases the risk of falling, which can lead to severe fractures and entry into physical dependence with a significant loss of quality of life and a cost to society, especially in a bone frail individual.

Osteoporosis and sarcopenia have common pathophysiological factors, including hormonal imbalance, increased inflammatory cytokine activity, nutritional changes, and physical impairment [8]. Common characteristics of these two pathologies have led to the description of a geriatric syndrome, called osteosarcopenia. This syndrome is associated with a higher risk of adverse outcomes in older people, such as loss of independence, falls, fracture risk, and institutionalization [9].

While this geriatric syndrome is emerging, the interactions between body compartments remain imperfectly understood. Several studies have shown that body fat mass is the main predictor of BMD in the elderly [10,11], while other studies suggest opposite results with lean body mass being significantly associated with BMD [12,13,14]. In contrast, physical activity is identified as an important factor in the regulation of body composition [1]. Physical activity is essential to both strengthen bone capital [15] and preserve muscle mass and function [16], and limit the risk of falls [17,18].

The main purpose of this study was to investigate the relationship between BMD and body composition (LM, FM) as related to physical activity level in women aged 75 or older. Secondary objectives were to describe the characteristics of patients (including body composition) suffering from osteoporosis and osteosarcopenia. 

## 2. Materials and Methods

### 2.1. Study Population

Between 1 January 2016 to 31 December 2019, this observational cross-sectional monocentric study was proposed to women aged 75 or older, addressed to the inner-city University Hospital of Nantes (Bellier Hospital, France) to perform a biphoton absorptiometry. Prior to any data collection, the investigator presented the interest of the study and ensured that an oral non-opposition was recorded in the file. For this study, a female population was studied, considering the significant differences between men and women in musculoskeletal aging, and that there are more published studies on bone aging in women.

The exclusion criteria were (i) contraindications to absorptiometry (behavioral disorders that may disturb the proper conduct of the examination, pain disturbing prolonged supine decubitus), (ii) patients under guardianship or trusteeship, and (iii) those unable to respond to a physical activity hetero-questionnaire. 

### 2.2. Baseline Data

The variables of interest were as follows: age, weight, height, body mass index (BMI), and moderate malnutrition defined by a BMI under 21 kg/m^2^ according to French guidelines [19]; living place (home vs. nursing home); patient addressed to biphoton absorptiometry by the geriatric outpatient platform or following hospitalization; serum albumin (g/L), serum 25-OH vitamin D (ng/mL); cognitive status, assessed by the Mini-Mental Scale examination (MMSE), ranging from 0 to 30 [20]; patient autonomy and dependence, assessed by the scales of Activities of Daily Living of Katz (ADL) and Instrumental Activities of Daily Living of Lawton (IADL); and morbidity, assessed by the Charlson comorbidity index (CCI) score.

For the purpose of this analysis, some scores were dichotomized at standard cut-off points which were defined according to current literature: MMSE < 24 points (cognitive impairment), and vitamin D deficiency if serum 25-OH vitamin D < 30 ng/mL. 

The ADL scale was validated [21] to judge the patient’s state of functional status (personal hygiene, dressing, using the toilet, locomotion, continence, and eating). The Instrumental Activities of Daily Living of Lawton (IADL) was validated [22] to assess the degree of dependence on 4 practical activities of daily living (using the telephone, means of transport, taking medication, and managing the budget). The ADL ranges from 0 (severe functional impairment) to 6/6 (full functional autonomy), and the IADL from 0 (dependency) to 4/4 (instrumental autonomy). The CCI score ranges from 0 to 33, and a high score is in favor of many comorbidities [23]. 

Patients also benefited from the evaluation of their physical performance assessed by a physical activity questionnaire: the Dijon physical activity score [24]. The research collaborator successively asked the 9 questions to the patient, then calculated a total score of 30 points, a high score reflecting a very active patient. 

Finally, physical performances were assessed for each patient by the hand grip strength, the unipedal stance test, and the 4-meter gait speed. 

### 2.3. Body Composition with BIPHOTON Absorptiometry

Patients underwent a biphoton absorptiometry or DXA to evaluate the body composition compartments by distinguishing FM and LM, both for different body regions (head, trunk, and limbs), but also for the whole body. Baumgartner et al. [25] deduced an appendicular skeletal muscular mass index (ASMM), defined by the ratio: arm muscle mass in kg + leg muscle mass in kg/height in m^2^. 

The device used was a LINAR iDXA^®^ with enCORE™ software version 12.x, Windows-XP Professional 2008. The results of the DXA are operator dependent (importance of the correct positioning of the patient on the examination table during acquisitions); thus, all measurements were performed by the same technician. The enCORE™ software used a precision calculator that evaluated the risk of measurement error and calculated its variability with a 95% confidence interval. 

### 2.4. Study Outcomes

The primary outcome was the bone mineral density defined by femoral neck T-score on a biphoton absorptiometry or DXA.

The secondary outcomes were as follows: sarcopenia [4], as defined according to the European Consensus by a hand grip test lower than 16 kg associated with an ASMM lower than 5.5 kg/m^2^; severe sarcopenia was considered when those criteria were associated with a gait speed lower than 0.8 m/s; osteoporosis was evaluated by the femoral neck T-score lower than −2.5 SD according to French guidelines; osteopenia was considered for femoral neck T-scores between −1 and −2.5 SD [2,3]; osteosarcopenia [26], which associates low bone mass and low muscle mass as defined according to the above definitions. 

### 2.5. Statistics

The participants’ characteristics were described using percentages for categorical variables, and the means and standard deviations for continuous variables.

Univariable linear regression was used to identify factors (including FM, LM, and physical activity level) associated with femoral neck T-score changes. All baseline variables with a *p*-value < 0.20 in univariable analysis and all the variables already known to be confounding factors were included in a step-by-step multivariable linear regression. The sample size required to obtain a power of 0.8 was measured at 79 patients (α = 0.05 and f^2^ = 0.15 (intermediate effect size)). Regression coefficients and confidence intervals (CI) were presented in the charts.

For the osteoporotic and osteosarcopenic patients’ characteristics, the continuous data were expressed as either mean (with standard deviation) and were compared using Student’s t-test or Mann–Whitney U-test depending on normality (Shapiro–Wilk test). For categorical variables, data were presented as frequencies and percentages and were compared using the Chi-squared test. A *p*-value less than 0.05 was considered statistically significant. All statistics were performed using the R software (version 3.4.3).

## 3. Results

### 3.1. Population Characteristics

Between 1 January 2016 and 31 December 2019, a total of 101 patients were included in the inner-city University Hospital of Nantes. Table 1 shows the detailed baseline characteristics of the whole population. Patients were women of Caucasian ethnicity, with a mean age of 84.8 years old (±4.9). The majority of them were living at home (92%) and were mostly independent, i.e., with low ADL and IADL scores (mean ADL = 5.2 (±1.0), and mean IADL = 2.8 (±1.2)). The mean BMI was 24.8 kg/m^2^ (±5.6); 29 patients were under 21 kg/m^2^ (28.7%), 32 were between 21 and 25 kg/m^2^ (31.7%), 22 were between 25 and 30 kg/m^2^ (21.8%), and 18 were over 30 kg/m^2^ (17.8%), with only 5 women having a BMI greater than 35 kg/m^2^ (5.0%). According to French guidelines (17), moderate malnutrition was considered for 29 patients (28.7%). A 25-OH Vitamin-D deficiency was found in 61 patients (60.4%). The mean MMSE score was 23.7 (±4.9) and 43 patients had cognitive impairment (42.6%). Patients were mildly comorbid, with a mean CCI score of 3.1 (±1.8). The majority of patients had a DXA examination during hospitalization (*n* = 79 (78%) vs. *n* = 21 (22%) of outpatients). Of the 79 patients hospitalized, 67% were being followed up for an ongoing severe fracture (vertebral or femoral). Of the total population, this represented 53 patients (52.5%) with a severe fracture at baseline.

Patients reported a preserved physical activity level, according to the Dijon physical activity level questionnaire, with a mean score of 21.5 (±5.3). However, the mean 4-meter gait speed and hand grip strength were impaired, respectively, of 0.62 m/s (±0.23) and of 10.3 kg (±6). The balance abilities were impaired, considering the observed failure in the unipedal stance under 5 s in 71 (70%) patients.

### 3.2. Study Outcomes

The femoral neck T-score was low with a mean of −2.23 SD (±1.05), resulting in a high prevalence of osteoporosis (72%) (Table 1). Osteopenia was found in 16 patients (16%). Of the 101 patients, only 12 had a BMD that did not satisfy the criteria for either osteopenia or osteoporosis. According to EWGSOP2 criteria [4], 37% of patients were sarcopenic. The combination of these two diseases, called osteosarcopenia, was present in 34% of patients.

### 3.3. Factors Associated with the Femoral Neck T-Score 

In univariable linear regression analysis, a significant association was found between the femoral neck T-score and age (β = −0.37, 95% CI (0.09; 0.46), *p* < 0.01), BMI (β = 0.43, 95% CI (0.25; 0.58), *p* = 0.04), ADL (β = 0.29, 95% CI (0.09; 0.46), *p* < 0.01), albumin (β = 0.28, 95% CI (0.07; 0;47), *p* < 0.01), physical activity level according to the Dijon physical activity questionnaire (β = 0.29, 95% CI (0.09; 0.46), *p* < 0.01), FM (β = 0.41, 95% CI (0.23; 0.034), *p* < 0.001), ASMM (β = 0.25, 95% CI (0.05; 0.43), *p* = 0.017), and 4-meter gait speed (β = −0.33, 95% CI (−0.50; −0.14), *p* < 0.01) (Table 2). 

In multivariable linear regression analysis, a significant association remained between the femoral neck T-score and age (β = −0.055, 95% CI (−0.075; −0.034), *p* = 0.009), and FM (β = 0.023, 95% CI (0.012; 5.99), *p* = 0.037) (Table 3).

### 3.4. Factors Associated with Osteoporosis

Table 4 shows characteristics of patients according to osteoporosis diagnosis. Regarding the body composition, osteoporotic patients had significantly lower LM and ASMM than non-osteoporotic patients (respectively, 34.1 kg (±4.4) vs. 37.0 kg (±4.9), *p* = 0.015); 5.7 kg/m^2^ (±0.9) vs. 6.2 kg/m^2^ (±0.8), *p* < 0.01). The FM was significantly lower in osteoporotic patients (18.2 kg (±8.5) vs. 28.1 kg (±11.0), *p* < 0.001).

Patients with osteoporosis had a significantly higher mean age than non-osteoporotic patients (85.8 (±4.9) vs. 82.3 (±4.1) years old, *p* < 0.01). The BMI was significantly lower in osteoporotic patients (23.5 kg/m^2^ (±5.1) vs. 28 kg/m^2^ (±5.7), *p* < 0.001). Malnutrition was found to be more prevalent in the osteoporotic population (*n* = 27 (37%) vs. *n* = 2 (7.1%), *p* < 0.01). 

### 3.5. Factors Associated with Osteosarcopenia

Table 5 shows characteristics of patients according to osteosarcopenia diagnosis. Osteosarcopenic patients had a significantly lower FM (16.2 kg (±6.8) vs. 23.1 kg (±10.8), *p* < 0.001). The BMI was also lower (20.7 kg/m^2^ (±2.8) vs. 26.7 kg/m^2^ (±5.6), *p* < 0.001). Malnutrition was more common in osteosarcopenic patients (*n* = 17 (53%) vs. *n* = 12 (17%), *p* < 0.001).

## 4. Discussion

Despite the controversial association between BMD and body composition (FM, LM) [10,11,12,13,14], the concept of osteosarcopenia emerged recently because both diseases share common causes and consequences [9]. At present, the determinants of the interactions are not fully described. This study was designed to explore these interactions. 

In this study, the femoral neck T-score was significantly associated with FM. The osteoporotic patients had significantly lower LM, FM, and BMI than non-osteoporotic patients. Osteosarcopenic patients had significantly lower FM and BMI than non-osteosarcopenic patients. 

The association between femoral neck T-score and FM found suggests a protective effect on the risk of osteoporosis. The currently available literature is controversial regarding the influence of FM on BMD. In 2016, He et al. [27] studied a population of 17,891 female and male subjects under 65 years of age. Their study highlighted a positive association between BMD and LM but a negative association with FM. Other studies have found an association between LM and BMD regardless of age and physical activity level, but not FM [12,13]. However, in contrast to young subjects, a significant association of BMD with FM was found in women over 65 years of age. In a 2017 study [28], based on the Pro-Saude study cohort in Brazil (100 premenopausal women versus 166 postmenopausal women), a negative association was found between BMD and FM in pre-menopause, whereas it became positive in post-menopause. In fact, these contradictions could be explained by differences in modifications of muscle mass and FM with age between women and men. Indeed, Du et al. [29] showed that men over the age of 65 years old had a much greater loss of muscle mass than women. They also showed that, over this period, older women had a much more significant increase in FM than men. In women, changes in body composition correspond mainly to an increase in body fat, which could explain a more predominant role of FM. In light of these elements, the association between BMD and body fat is consistent with results in the literature concerning older postmenopausal women. A suggested pathophysiological explanation for the protective role of FM on BMD is developed in the literature. Indeed, during this period, fat tissue is responsible for estrogen production and maintains estrogen at a certain level after menopause. Therefore, as a compensatory mechanism, FM increases significantly during perimenopause and is maintained at a high level after, limiting the estrogenic deficiency and its consequences on the bone [29]. 

In this study, there was not a significant association between BMD with LM and physical activity. However, osteoporotic patients had significantly lower LM and ASMM. Finally, 25-OH vitamin D level was not associated with BMD changes and was not significantly lower in osteoporotic or osteosarcopenic patients. The interactions between muscle and bone are well described. They interact through their proximity and locomotor function. In addition, these two tissues modulate their development according to a paracrine and endocrine dialogue [30,31,32]. Osteoporosis and sarcopenia would have common pathophysiological factors, including hormonal imbalance, increased inflammatory cytokine activity, nutritional changes, and physical impairment [8]. For example, the rapid decline in estrogens after menopause and the gradual decrease of androgens in older men lead to a reduction in muscle and bone mass, with an increased risk of falls and fractures [33]. However, many other tissue-specific factors released by muscles also modulate bones, like insulin-like growth factor-1 (IGF-1), Il-6, and myostatin. Some of these factors are involved in the pathogenesis of sarcopenia. 25-OH vitamin D is also a modulator of osteogenesis and anabolism, through the presence of multiple vitamin D receptors on the surface of these tissues. [30,34]. We assume that our results could be explained by a lack of power and a bias associated with the study population selection. In this study, the older women had a significant alteration in terms of strength and function, whereas the study population reported a preserved level of physical activity. This inconsistency could be explained by the temporality of the study: the majority of patients were hospitalized, which could induce an alteration in physical performance. It could also be explained by a reporting bias. Finally, vitamin D did not appear to be an influencing factor in our study, although the reverse was expected. Beyond the lack of power, the explanation is related to the fact that our population had on average very low vitamin D deficiency.

In this study population, the prevalence of osteosarcopenia was important, found in 34% of the included patients. Osteosarcopenic patients had lower BMI and FM than non-osteosarcopenic patients. In addition, the prevalence of malnutrition was important in osteosarcopenic patients. In the literature, osteosarcopenia is described as a lipotoxic disease [9]. Indeed, the progressive decline in bone and muscle mass is associated with fat infiltration [35]. Marrow adipose tissue infiltrates the marrow cavity and muscle fat infiltrates intra- and inter-fiber spaces. The presence of high levels of fat is associated with an increase level of apoptosis and autophagy. Marrow and muscle adipocytes product free fatty acids, that create local toxicity. These adipocytes also secrete adipokines (adiponectin, leptin) that reach the circulation and have a systemic effect on bone and muscle mass and function [36]. Several explanations can be given to highlight our results being inconsistent with the idea of a lipotoxicity. It is interesting to observe the distribution of BMI in our study population. The majority of patients had a BMI between 21 and 30 kg/m^2^ (*n* = 54 (53.5%)), 12.9% of patients had a stage 1 obesity (BMI between 30 and 35 kg/m^2^), and four women presented with stage 2 obesity. This population was not severely obese, and predominantly overweight. Being overweight has been described as a protective factor against osteoporosis [37]. These effects of weight on BMD are explained by mechanical stresses and forces exerted on bone, which stimulate the process of bone formation through the action of osteoblasts [13]. Furthermore, BMI does not give a perfect idea of FM distribution. For example, this finding was made in the area of cardiovascular risk. Thus, a systematic review showed a decrease in mortality for BMI between 25 and 30 kg/m^2^ and no increase in mortality for BMI between 30 and 35 kg/m^2^ compared to BMI between 18 and 25 kg/m^2^. This is known as the obesity paradox [38]. This paradox is partially explained by the fact that BMI does not reveal the accumulation of FM at the visceral level. Android obesity is responsible for significant lipotoxicity with the secretion of pro-inflammatory factors such as cytokines and adipokines [39]. Do variations in fat distribution induce different effects on muscle and bone tissue, as described in the obesity paradox? In this hypothesis, the FM measurement by DXA does not make a report on fat distribution. In addition, DXA does not identify intramuscular lipid infiltration and overestimates muscle mass in overweight or obese patients [35]. Another explanation for our results is the interaction with malnutrition, an important confounding factor partially taken into account. Malnutrition (or even a state of cachexia with a chronic disease) induces an overall loss of body mass. This factor is recognized as a risk factor of osteoporosis and sarcopenia [3,4,40]. It might not be the fact of having excess fat that is protective, but rather having an overall loss of body mass, potentially reflecting a state of malnutrition or cachexia. Finally, this study focused on a population of older women. However, the literature seems to show significant differences between men and women in the interactions between body compartments. In older men, LM would appear to contribute more to improving BMD, while in older women, FM and BMD would have a greater association [41].

This study had limitations. First, several confounding factors that could introduce a bias in the results were not identified including smoking, alcohol consumption, comorbidities such as endocrinopathies (increasing the risk of osteoporosis), and treatments (such as corticosteroid therapy) [2,3]. Indeed, it was not possible to accurately record the present or past smoking status or alcohol consumption due to a reporting bias. In addition, since the rheumatological follow-up was performed out-of-hospital, we did not have access to data on anti-osteoporotic treatments at the moment of inclusion. Second, the use of a physical activity questionnaire induces a self-report bias. This is demonstrated by the inconsistency between the score results and the objectified physical performance. Third, the study population was predominantly hospitalized patients who were referred to DXA in a post-fracture context. Thus, they were more at risk of osteoporosis and sarcopenia, which leads to a lack of representativeness. In addition, the fracture context can induce systemic inflammation, which can influence all body compartments negatively [42,43].

For the future, and in order to confirm and better understand the interactions between BMD, LM, and FM, it would be appropriate to conduct a general population study including both elderly men and women. Above all, it would be interesting to develop studies with other markers analyzing FM, such as anthropometric variables, making it possible to elaborate on the fat distribution (waist circumference?). Moreover, biomarkers could be added to determine the inflammation level based on FM location. Furthermore, DXA has several limitations to explore different compartments. Indeed, it is not easily feasible in routine care, and it does not allow for the visualization of fat distribution. Other complementary exploration should be discussed, such as the development of ultrasound-based approaches for a quantitative and qualitative characterization of muscle composition, for example [44].

## 5. Conclusions

In conclusion, musculoskeletal alterations are a major contributor to disability among older people. Thus, osteoporosis and sarcopenia are linked, with an increased risk of falls, fractures, loss of independence, and institutionalization. The concept of osteosarcopenia is growing, now being described as a geriatric syndrome. The importance and role of fat mass on bone and muscle are still uncertain among elderly women.

Our study may suggest a protective effect of fat mass. This hypothesis is supported from a pathophysiological point of view by the production of estrogens by adipose tissue to compensate the effects of menopause, or by the mechanical stresses of fat mass on bone tissue. 

However, adipocytes are also known to produce pro-inflammatory factors that are harmful to the bone and muscle. Several hypotheses are then proposed. FM may have a different role depending on its location, with induction of different local and systemic inflammatory toxicity depending on the location. It seems important to be able to develop tools to assess the distribution of fat mass, associated with the development of biomarkers of inflammation. DXA seems insufficient in this approach, which opens the way for other approaches such as the ultrasound.

## Figures and Tables

**Table 1 nutrients-13-01994-t001:** Population characteristics.

	Total (*n* = 101)
Patients’ characteristics	
Age (years), mean ± SD	84.8 ± 4.9
BMI (kg/m^2^), mean ± SD	24.8 ± 5.6
<18 kg/m^2^, *n* (%)	7 (6.9)
18–21 kg/m^2^, *n* (%)	22 (21.8)
21–25 kg/m^2^, *n* (%)	32 (31.7)
25–30 kg/m^2^, *n* (%)	22 (21.8)
30–35 kg/m^2^, *n* (%)	13 (12.9)
>35 kg/m^2^, *n* (%)	5 (5.0)
Malnutrition, *n* (%)	29 (28.7)
MMSE, mean ± SD	23.7 ± 4.9
ADL, mean ± SD	5.2 ± 1.0
IADL, mean ± SD	2.8 ± 1.2
Charlson comorbidity index, mean ± SD	3.1 ± 1.8
Albumin (g/L), mean ± SD	35.0 ± 4.9
25-OH Vitamin D (ng/mL), mean ± SD	25.1 ± 14.1
Inclusion during hospitalization	79 (78)
For a vertebral fracture	33 (42)
For a femoral fracture	20 (25)
For another reason	26 (33)
Outpatient	21 (22)
Physical performances	
4-meter gait speed (m/s), mean ± SD	0.62 ± 0.23
Hand grip (kg), mean ± SD	10.3 ± 6.0
Unipedal stance >5 s, *n* (%)	30 (30)
Physical activity level, mean ± SD	21.5 ± 5.3
DXA measurements	
Femoral neck T-score (SD), mean ± SD	−2.23 ± 1.05
Lean mass (kg), mean ± SD	34.9 ± 4.7
Fat mass (kg), mean ± SD	20.9 ± 10.2
ASMM (kg/m^2^), mean ± SD	5.8 ± 0.9
Musculoskeletal alterations	
Osteopenia, *n* (%)	16 (16)
Osteoporosis, *n* (%)	73 (72)
Osteosarcopenia, *n* (%)	33 (34)
Sarcopenia, *n* (%)	37 (37)

SD indicates standard deviation; BMI, body mass index; MMSE, Mini-Mental Scale examination; ADL, Activities of Daily Living; IADL, Instrumental Activities of Daily Living; physical activity level according to the Dijon physical activity questionnaire; ASMM, appendicular skeletal muscle mass.

**Table 2 nutrients-13-01994-t002:** Univariable analysis of factors associated with the femoral neck T-score in women patients aged ≥75 years.

Variables	β (95% CI) *	*p* Value **
Lean mass	0.19 (−0.02; 0.37)	0.072
Fat mass	0.41 (0.23; 0.56)	**<0.001**
ASMM	0.25 (0.05; 0.43)	**0.017**
Physical activity level	0.29 (0.09; 0.46)	**<0.01**
Age	−0.37 (0.09; 0.46)	**<0.01**
BMI	0.43 (0.25; 0.58)	**<0.001**
Charlson comorbidity index	−0.19 (−0.38; 0.01)	0.059
Albumin	0.28 (0.07; 0.47)	**<0.01**
25-OH Vitamin D	0.12 (−0.10; 0.32)	0.29
4-meter gait speed	−0.33 (−0.50; −0.14)	**<0.01**
Hand grip	−0.02 (−0.23; 0.18)	0.83

* β (95% IC): linear regression coefficient β (95% confidence interval). ** *p* value: significant difference if *p*-value < 0.05 (highlighted in bold). BMI, body mass index; MMSE, Mini-Mental Scale examination; physical activity level according to the Dijon physical activity questionnaire; ASMM, appendicular skeletal muscle mass.

**Table 3 nutrients-13-01994-t003:** Multivariable analysis of factors associated with the femoral neck T-score in women patients aged ≥75 years.

Variables	β (95% CI) *	*p* Value **
Age	−0.06 (−0.08; −0.03)	**0.009**
Physical activity level	0.04 (0.02; 0.06)	0.053
Fat mass	0.02 (0.01; 0.03)	**0.037**
ASMM	0.19 (0.07; 0.32)	0.121

* β (95% IC): linear regression coefficient β (95% confidence interval). ** *p* value: significant difference if *p*-value < 0.05 (highlighted in bold). ASMM, appendicular skeletal muscle mass.

**Table 4 nutrients-13-01994-t004:** Patients’ characteristics according to osteoporosis diagnosis (*n* = 101).

Total (*n* = 101)	Osteoporosis		*p* Value *
	No (*n* = 28)	Yes (*n* = 73)	
Patients’ characteristics			
Age (years), mean ± SD	82.3 (±4.1)	85.8 (±4.9)	**<0.01**
BMI (kg/m^2^), mean ± SD	28.3 (±5.7)	23.5 (±5.1)	**<0.001**
Malnutrition, *n* (%)	2 (7,1)	27 (37)	**<0.01**
Charlson comorbidity index, mean ± SD	3.4 (±1.9)	2.9 (±1.8)	0.27
Albumin (g/L), mean ± SD	34.9 (±5.0)	35.1 (±4.9)	0.86
25-OH Vitamin D (ng/mL), mean ± SD	25.7 (±13.2)	24.9 (±14.5)	0.77
Physical performances			
4-meter gait speed (m/s), mean ± SD	0.60 (±0.21)	0.62 (±0.24)	0.65
Hand grip (kg), mean ± SD	11.6 (±7.2)	9.8 (±5.5)	0.15
Unipedal stance >5 s, *n* (%)	6 (21%)	24 (33%)	0.26
Physical activity level, mean ± SD	22.2 (±4.2)	21.2 (±5.6)	0.59
DXA measurements			
Lean mass (kg), mean ± SD	37.0 (±4.9)	34.1 (±4.4)	**0.015**
Fat mass (kg), mean ± SD	28.1 (±11.0)	18.2 (±8.5)	**<0.001**
ASMM (kg/m^2^), mean ± SD	6.2 (±0.8)	5.7 (±0.9)	**<0.01**

* *p* value: significant difference if *p*-value < 0.05 (highlighted in bold). SD indicates standard deviation; BMI, body mass index; physical activity level according to the Dijon physical activity questionnaire; ASMM, appendicular skeletal muscle mass.

**Table 5 nutrients-13-01994-t005:** Patients’ characteristics according to osteosarcopenia diagnosis (*n* = 101).

Total (*n* = 101)	Osteosarcopenia		*p* Value *
	No (*n* = 68)	Yes (*n* = 33)	
Patients’ characteristics			
Age (years), mean ± SD	84.9 (±5.0)	84.6 (±4.7)	0.8
BMI (kg/m^2^), mean ± SD	26.7 (±5.6)	20.7 (±2.8)	**<0.001**
Malnutrition, *n* (%)	12 (17)	17 (53)	**<0.001**
Charlson comorbidity index, mean ± SD	3.3 (±1.8)	2.6 (±1.7)	0.091
Albumin (g/L), mean ± SD	35.1 (±4.8)	34.9 (±5.1)	0.86
25-OH Vitamin D (ng/mL), mean ± SD	25.4 (±13.0)	24.5 (±16.2)	0.8
Physical performances			
Unipedal stance >5 s, *n* (%)	15 (22)	15 (47)	**0.01**
Physical activity level, mean ± SD	21.2 (±5.6)	22.1 (±4.6)	0.4
DXA measurements			
Fat mass (kg), mean ± SD	23.1 (±10.8)	16.2 (±6.8)	**<0.001**

* *p* value: significant difference if *p*-value < 0.05 (highlighted in bold). SD indicates standard deviation; BMI, body mass index; physical activity level according to the Dijon physical activity questionnaire.

## Data Availability

Data collection was encrypted in an Excel database, in accordance with current regulations. The Excel file was created by the inner-city University Hospital of Nantes and was hosted via its computer network. A coding system allowed data to be transmitted to an investigator anonymously.

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
