# Peer review of "Association between Bone Mineral Density and Fat Mass Independent of Lean Mass and Physical Activity in Women Aged 75 or Older"

_nutrients, 2021, doi:10.3390/nu13061994_

Round 1

Reviewer 1 Report

The manuscript by Mathieu et al. provides an interesting exploration between the relationships between body composition and bone mineral density in elderly women. This manuscript provides new data to the field and should be of interest to the journal's readership. However, several items require attention. These are outlined below. 

The authors should state the ethnicity of the study population. The authors should also provide some rationale for not assessing or at least collecting additional baseline characteristics that may confound the findings. For example, smoking status or drug use (i.e., steroid use, anti-osteoporosis drugs) are not considered. 

If possible it would be interesting to further stratify the results into three BMD categories: 1) Normal; 2) osteopenic; 3) osteoporotic. 

Using BMI as the sole measure of malnutrition is not appropriate. Were any serum samples and/or dietary recalls collected to assess other measures of malnutrition? 

In the limitation section the authors state that the study population largely consists of patients referred to DXA after fracture. This is a critical limitation that needs to be further explored and explained. Fracture induces systemic bone loss and may confound the findings. Additional information regarding fracture location, severity, etc need to be provided. 

Reviewer 2 Report

The manuscript demonstrated the relevance of fat mass and BMD in women aged 75 or older. The overall are interested, however, the following points should be considered.

  1. Method:  The sample size calculation before start of the study was not described in the method section. Please mention this.
  2. The number of participants in the control (n=28) and patient (n=73) groups are quite different. Furthermore, the number of the patients with osteoporosis and osteosarcopenia  are the same (n=73, Table 4 and 5). Is this correct?  
  3. Please mention the Declaration of Helsinki in the ethical consideration section.
  4. Results: Why did not the authors analyze the body weight, but not BMI?  I think body weight might be the most effective factor to BMD.
  5. How the malnutrition of the subjects was assessed?
  6. Discussion: For the explanations of the  relevance of FM and BMD in the elder women, the authors mentioned  the possibility of estrogen production from fat tissue, adipokine and malnutrition. However, none of these were shown in the Result section.  Please describe these in the discussion.
  7. Please discuss the relationship between serum 25(OH)D level and BMD in the subjects. 
  8. The conclusions are not clear. 

Reviewer 3 Report

The proposed study aimed to elucidate the relationship between bone mineral density (BMD) and body composition in women older than 75 years. Therefore, an observational crosssectional monocentric study was proposed and a biphoton absorptiometry was performed. 
Some  minor comments were raised during the revision process:Line 32: there is a minor mistake, it "The decrease..:"

  1. Line 34: Either "decrease in BMD" or "decreased BMD"

Round 2

Reviewer 1 Report

The authors have adequately addressed my concerns.